# Possible Roles of tRNA Fragments, as New Regulatory ncRNAs, in the Pathogenesis of Rheumatoid Arthritis

**DOI:** 10.3390/ijms22179481

**Published:** 2021-08-31

**Authors:** Satoshi Yamasaki, Munetoshi Nakashima, Hiroaki Ida

**Affiliations:** 1Division of Rheumatology, Kurume University Medical Center, 155-1 Kokubu-machi, Kurume 839-0863, Fukuoka, Japan; mnakashima@med.kurume-u.ac.jp; 2Division of Respirology, Neurology and Rheumatology, Department of Medicine, Kurume University School of Medicine, Kurume 830-0011, Fukuoka, Japan; ida@med.kurume-u.ac.jp

**Keywords:** non-coding RNA, tiRNA, angiogenin, ribonuclease inhibitor 1, Schlafen 2, stress granule, terminal oligoguanine motif, guanine quadruplexes, Y-box binding protein 1

## Abstract

Understanding the pathophysiology of rheumatoid arthritis (RA) has led to the successful development of molecule-targeted drugs for the treatment of RA. However, some RA patients are refractory to these treatments, suggesting that the pathological mechanism of the disease is not entirely understood. Genome and transcriptome analysis is essential for understanding the unknown pathophysiology of human diseases. Rapid and more comprehensive gene analysis technologies have revealed notable changes in the expression of coding RNA and non-coding RNA in RA patients. This review focuses on the current state of non-coding RNA research in relation to RA, especially on tRNA fragments. Interestingly, it has been found that tRNA fragments repress translation and are antiapoptotic. The association between tRNA fragments and various diseases has been studied, and this article reviews the possible role of tRNA fragments in RA.

## 1. Introduction

Rheumatoid arthritis (RA) is characterized by inflammatory changes in the synovial tissue and the destruction of the bone and cartilage in the joints. The genetic background and acquired factors such as smoking, periodontal disease, and microbial flora are important in the pathogenesis of RA [1,2,3]. Immune disorders of both the adaptive and the innate immune systems cause chronic inflammation. The appearance of autoantibodies, such as rheumatoid factor (RF) and anticitrullinated peptide antibody (ACPA), has been associated with these immune disorders. Inflammatory cytokines such as tumor necrosis factors-α (TNFα) and interleukin-6 (IL-6) contribute to synoviocyte and lymphocyte activation. These advances in understanding the pathophysiology of RA have resulted in the development of revolutionary treatment strategies. Biologics, such as monoclonal antibodies and receptor proteins, effectively neutralize pathogenic inflammatory cytokines (TNFα [4,5,6], IL-6 [7,8]) or block costimulatory molecules on antigen-presenting cells [9]. Janus kinase inhibitors (JAK inhibitors) have a therapeutic effect for RA comparable to the effect of biologics [10,11]. Treatment with these drugs significantly ameliorates RA and has consolidated our understanding of RA [12].

Despite significant therapeutic advances, unsolved problems remain. First, some patients do not achieve remission with the latest treatments. These are defined as “difficult-to-treat (D2T) RA” cases by the European League Against Rheumatism (EULAR) [13]. It is necessary to design specific therapeutic strategies for such patients.

Once RA is in remission, the goal is to achieve drug-free remission. Remission of RA only means suppressing inflammation, and, currently, there is no clear pathophysiological distinction between drug-free remission and cure. In fact, defining a cure for RA is challenging. However, if the distinction between drug-free remission and cure is clearly identified, drugs can be confidently discontinued. Technologies that can assist in making this distinction will help to elucidate the pathophysiology of RA in more detail, leading to the development of novel treatments aimed at a complete cure [14].

Quantifying the expression of messenger RNA (mRNA) has been the primary analysis method in RA research. This is because the expression levels of genes such as cytokines and proteases are clinically important for RA development and progression. In recent years, RNA sequencing (RNA-seq) has been used for transcriptome analysis of RA blood samples [15]. In addition, studies assessing changes in the transcriptome before and after a treatment have revealed the genes whose expression needs to be regulated to achieve remission [16,17]. These studies corroborate our knowledge of RA, clarify the mechanism of action of therapeutic agents, and sometimes reveal unknown pathophysiological features of RA [18]. Comprehensive research using innovative analysis has also revealed that the aberrant regulation of non-coding RNA (ncRNA) in RA patients is more frequent than expected. This review describes the current state of ncRNA research in relation to RA, focusing on tRNA fragments and their relevance to RA pathophysiology.

## 2. ncRNAs and Rheumatoid Arthritis

The complete analysis of the human genome revealed that coding sequences occupy only 1.5% of the genome, and about 25% of the genome is transcribed into pre-mRNAs with introns [19,20]. ncRNAs have attracted attention as transcripts that explain the role of the remaining 75% of the genome. Ribosomal RNA (rRNA) and transfer RNA (tRNA) had already been discovered as ncRNAs of biological significance in the 1950s. RNA interference (RNAi), described by Fire and Mello in 1998, opened up a new world of ncRNAs [21,22,23]. Advances in sequence information analysis have accelerated the discovery of ncRNAs [24,25], and the aggregation of this information has facilitated ncRNA research and has been applied to the study of human diseases [26,27].

Classical ncRNAs such as rRNA, tRNA, small nuclear RNAs, and small nucleolar RNAs are usually constitutively expressed and are regarded as housekeeping ncRNAs, whereas other ncRNAs are called “regulatory ncRNAs” because they regulate transcription, RNA processing, and translation [28,29]. The most studied regulatory ncRNAs are microRNAs (miRNAs). Research on miRNAs started with the discovery of the first miRNA, *lin-4*, in *Caenorhabditis elegans* in 1993 [30]. RNA interference (RNAi), a sequence-specific mRNA suppression phenomenon induced by double-stranded RNA, was discovered in 1998 [21,22]. After miRNA is incorporated into Argonaute proteins to form an RNA-induced silencing complex (RISC) in the cytoplasm, RISC suppresses its target gene expression by binding to the 3′ untranslated region (3′ UTR) of the target mRNA and induces mRNA destabilization and translational repression [31].

In 2008, three groups reported aberrant miRNA expression in the peripheral blood or synovium of RA patients [32,33,34]. MiRNAs recognize complementary sequences in the 3′ UTR of mRNAs and suppress the expression of target genes. Therefore, reduced miRNA expression leads to the enhanced expression of target mRNAs associated with RA pathology. Two studies published in 2008 found the reduced expression of miR-146 in patients with RA. This finding is reasonable, because miR-146 targets *IRAK1* and *TRAF6* mRNAs, which are downstream signaling molecules of pivotal inflammatory cytokines, i.e., TNFα and interleukin-1β (IL-1β). Notably, miRNAs can survive in the human blood, urine, saliva, and other body fluids because they are encapsulated in exosomes, vesicles covered with a lipid bilayer 20–100 nm in diameter [35]. Encapsulation of miRNAs in exosomes facilitates their cell-to-cell transport, allowing the regulation of gene expression in target cells, similar to cytokines. Therefore, miRNAs are receiving more attention in clinical medicine, including RA research [36].

Long non-coding RNAs (lncRNAs) are ncRNAs consisting of 200 nucleotides or more. A huge number of lncRNAs have been discovered, which already exceed the number of known coding RNAs. Although the function of lncRNAs has not fully been elucidated, it is known that they regulate the expression of target genes through epigenetic, transcriptional, or splicing regulation [37]. Since 2014, there have been reports of various lncRNAs in RA [38].

So far, RA-related ncRNAs have been briefly reviewed. Many other regulatory ncRNAs have been reported in addition to miRNAs and lncRNAs [39]. These ncRNAs are associated with many diseases and are integrated into more complex networks. While these reports are very interesting, as they provide clues to elucidate the unknown pathological mechanism of RA, it is challenging to identify specific ncRNAs that can be useful as therapeutic targets. For example, miR-223 suppresses osteoclasts in humans [40] but has been reported to promote osteoclastogenesis in mouse arthritis models [41]. These conflicting results may be due to the off-target effects of miRNAs or may be the result of secondary gene regulation by miRNA target genes [42]. In any case, intense research is necessary to discover specific ncRNAs that can be considered as therapeutic targets.

The world of ncRNAs is further complicated by the fact that tRNAs, which were previously thought to be housekeeping ncRNAs, are transformed into regulatory ncRNAs by fragmentation. The following sections describe tRNA-derived, stress-induced small RNAs (tiRNAs). As the name implies, they are cleaved fragments of tRNA. A recent study found that tiRNAs were altered in the plasma of RA patients [43]. Whether changes in tiRNAs are the cause or the result of RA pathophysiology has not yet been elucidated. However, tiRNAs have the potential to be important players in RA pathophysiology.

## 3. Discoveries of New tRNA Functions

RNAs consist of 76–93 nucleotides transcribed by RNA polymerase Ⅲ. A transcribed pre-tRNA becomes a mature tRNA by a multi-step process. RNase P and RNase Z remove the 5′ leaders and 3′ trailers from pre-tRNAs, respectively. Introns are spliced out, and the nucleotides are modified. The process is completed by adding a CCA tail to the 3’ terminal, which acts as a site for amino acid charging. tRNA has a four-leaf clover-like secondary structure consisting of an acceptor stem, a dihydrouridine (D) stem–loop, an anticodon stem–loop, and a TψC stem–loop [44,45].

tRNAs are essential for translation, the process of protein synthesis. They transfer amino acids to a growing polypeptide chain on ribosomes. In addition to their classical functions, recent studies have revealed new functions for tRNAs. Interestingly, tRNAs can exert an antiapoptotic function. When cytochrome *c* is released from the mitochondria during cellular stress or DNA damage, it binds to Apaf-1 and activates the caspase cascade required for apoptosis. However, binding of tRNA to cytochrome *c* in the cytoplasm inhibits apoptosis [46,47,48].

Translation involves three steps: initiation, elongation, and termination. The formation of a preinitiation complex composed of the 40S ribosomal subunit, eukaryotic initiation factor 3 (eIF3), eIF1A, and eIF2/GTP/methionyl initiator tRNA initiates translation. The preinitiation complex binds to a capped mRNA in association with eIF4E (cap-binding protein), eIF4A (RNA helicase), and eIF4G (scaffold protein). The complex scans the 5′ UTR until the initiation codon is found, then translation starts [49].

Environmental stress mainly affects the translation initiation step, causing general translation silencing [49,50]. Inactivation of the preinitiation complex by phosphorylation of eIF2α at serine 51 is one of the most potent translational inhibitors. This phosphorylation is mediated by four kinases activated by various types of environmental stress: (1) protein kinase RNA (PKR) is activated by double-stranded RNA produced during viral infection; (2) heme-regulated inhibitor kinase (HRI) is activated by hemin and by arsenite, an oxidative stress inducer; (3) PKR-like endoplasmic reticulum (ER) kinase (PERK) is activated as a result of the unfolded protein response; and (4) activated general control non-depressible-2 (GCN2) phosphorylates eIF2α by binding to the deacylated tRNA during amino acid starvation [49,50,51,52,53]. These stress-induced translational repressions are mediated by the phosphorylation of eIF2α, but the finding that low-dose arsenite inhibits translation even in cells bearing non-phosphorylatable eIF2α, in which serine 51 was mutated to alanine (Ser51Ala) [53], suggested the existence of a phospho-eIF2α-independent translation-inhibitory pathway. Is tRNA involved in phospho-eIF2α-independent translational inhibition?

Changes in the localization of tRNAs may hamper global translation because cytoplasmic tRNAs translocate into the nucleus under various types of stress [54,55,56,57]. tRNAs are cleaved by environmental stress, and changes in the quantity of tRNA may suppress global translation. Cleavage of tRNA was first reported as a starvation response in *Tetrahymena thermophila* [58], and similar phenomena occur in *Streptomyces coelicolor* [59], *Aspergillus fumigatus* [60], *Trypanosoma cruzi* [61], *Giardia lamblia* [62], phosphate-depleted *Arabidopsis thaliana* [63], and *Saccharomyces cerevisiae* under oxidative stress [64]. tRNA cleavage has also been suggested to contribute to stress-induced translational arrest in mammalian cells [65]. Sequence analysis revealed that tRNAs are cleaved at or near the anticodon loop, and the 3′ fragment of the cleaved tRNAs contains the CCA sequence but lacks tRNA introns. These are the characteristics of mature tRNAs. Therefore, these tRNA fragments are derived from mature tRNAs and not from pre-tRNAs. The important fact is that the appearance of tRNA fragments does not result in a robust depletion of mature tRNAs capable of inhibiting global translation. Moreover, PIWI-associated RNAs (piRNAs) contain similar tRNA-derived fragments [66]. These facts suggest that tRNA cleaved at the anticodon site may be associated with the ribonucleoprotein complex, and the complex may contribute to translational repression independent of phospho-eIF2α.

## 4. tRNA-Derived Stress-Induced Small RNAs (tiRNA) in Translational Control

DICER [67] and ELAC2/RNase Z [68,69] constitutively produce tRNA-derived ncRNAs. Secretory RNases, RNY1 in yeast [64] and angiogenin (ANG) in humans [70], cleave mature tRNAs within the anticodon loop under stress and produce tRNA fragments (of ~30 and ~40 nucleotides). The amount of stress-induced tRNA fragments is much lower than that of mature tRNAs, and translation probably cannot be suppressed by tRNA depletion. Oxidative stress, heat, or UV irradiation produce 5′ and 3′ tRNA fragments in mammalian cells. We defined these fragments as tRNA-derived stress-induced small RNAs (tiRNA) [71]. When endogenous tiRNAs were purified and transfected into U2OS cells, tiRNAs corresponding to the 5′ tRNA fragments (5′ tiRNA), but not to the 3′ tiRNA, suppressed protein synthesis. Similar results were obtained when MEFs (mouse embryonic fibroblasts) obtained from eIF2α mutant (Ser51Ala) mice were transfected with the 5′ tiRNA. This indicated that inhibition of protein synthesis does not require phosphorylation of eIF2α [71]. Recombinant ANG induces tiRNA production in mammalian cells and also inhibits global protein synthesis. Knockdown of endogenous ANG inhibited tiRNA production, which, in turn, released stress-induced translational repression. Knockdown of ribonuclease inhibitor 1 (RNH1) [72], an inhibitor of ANG, enhanced tiRNA production and accelerated stress-induced translational repression [70,71]. These results emphasize the role of ANG and tiRNA in stress-induced translational repression [73] (Figure 1).

Not all tiRNAs suppress protein synthesis to the same extent. In fact, 5′ tiRNA^Ala^ and 5′ tiRNA^Cys^ produced by cleaving tRNA^Ala^ and tRNA^Cys^, respectively, significantly suppressed protein synthesis [74]. Translation-suppressing tiRNAs have consecutive guanines at the 5′ end, which form the so-called the terminal oligoguanine (TOG) motif [75]. Some of these tiRNAs potentially form guanine quadruplexes, non-canonical structures consisting of G-rich sequences [76]. tiRNAs can inhibit translation by blocking the binding of the mRNA cap structure to the cap-binding complex, in coordination with the translation-suppressing Y-box binding protein 1 (YB-1) [74]. The TOG motif is important for the association of tiRNA with YB-1.

Notably, transfection of tiRNA into cells augments stress granule (SG) formation [77,78]. SGs are cytoplasmic foci consisting of 40S ribosomal subunits, mRNA, translation initiation factors, and RNA-binding proteins. SGs promote cell survival in stressful environments by classifying and storing untranslated mRNAs, RNA-binding proteins, and signaling molecules [79,80,81,82]. Even if tiRNAs cause global translational repression, SG formation will allow cells to quickly return to their original translational levels after the stress has been relieved. As expected, treating cells with recombinant wild-type ANG, which can produce tiRNAs, promotes SG formation, but mutant ANG does not [77,78]. It should be noted that, like tRNAs [46,47,48], tiRNAs also inhibit apoptosome formation by binding cytochrome *c* and promote cell survival [83]. Thus, tRNA fragments regulate translation, SG assembly, and apoptosis (Figure 1).

It is known that there are multiple pathways for the production of tRNA fragments. While the production of tiRNA is induced by various stresses, different types of tRNA fragments are also constantly produced. In contrast to tiRNAs, which are fragments of mature tRNAs, there are tRNA fragments containing the 5′-leader and 3′-trailer sequences that are characteristic of immature tRNAs. tiRNA is produced by cleavage in the anticodon loop by ANG, but some “non-tiRNA” tRNA fragments are cleaved in the D stem–loop or T stem–loop. It is thought that RNA-cleaving enzymes other than ANG, such as DICER and ELAC2/RNase Z, also contribute to the production of tRNA fragments [84]. Since such abundant types of tRNA fragments are produced depending on the situation, it is natural that tRNA fragments perform various functions. Although tRNAs were treated as housekeeping ncRNAs, these recent discoveries indicate that tRNA-derived fragments are new members of the regulatory ncRNAs [85].

## 5. Possible Roles of tiRNA in Rheumatoid Arthritis

Research on tRNA fragments in human diseases is on the rise [86,87,88,89]. Zhang and colleagues reported the altered expression of certain tiRNAs in lung adenocarcinoma compared with the surrounding normal lung tissue [90]. Whether the tRNA fragments present in cancer tissues are tiRNAs or another type of tRNA fragments still remains to be understood [91,92]. Reports showing the association of tiRNAs with immunity are also increasing. For example, virus infection has been shown to induce tiRNA production. A549 human alveolar Type II-like epithelial cells infected with human respiratory syncytial virus (RSV) produce abundant ANG-dependent tRNA-derived RNA fragments [93]. Public data regarding tRNA fragments have been released to facilitate analysis [94], and research on tRNA fragments is expected to expand in the future.

Although research on the tRNA fragments involved in RA has just started, several studies about ANG and RA have been reported, suggesting the involvement of tiRNAs in the disease. ANG was discovered as a molecule that promotes angiogenesis [95]. ANG has been intensively analyzed as a molecule involved in tumors and inflammation. In 1998, ANG was shown to be a liver-derived acute-phase protein, similar to serum amyloid A protein [96]. Consistent with these results, it was reported that the concentrations of ANG in synovial fluid were higher in RA patients than in osteoarthritis patients [97]. An inflamed joint can be the origin of ANG production because fibroblast-like synoviocytes (FLSs) also secrete ANG [97]. No RNH1 studies on RA have been conducted so far.

Since ANG is elevated in RA patients, it was expected that there would be an aberrant expression of tiRNAs in RA. Ormseth and colleagues found that tiRNAs, along with miRNAs, were significantly elevated in RA patients’ serum compared with the serum of healthy individuals [43]. Some of these tiRNAs have a G-rich sequence and the potential to suppress global translation, similar to 5′ tiRNA^Ala^ and 5′ tiRNA^Cys^ [74,75]. The elevation of tiRNAs in RA patients is interesting, but there is no answer to the essential question, “Is tiRNA the cause or the effect of RA?”.

TiRNAs have been reported in serum, spleen, bone marrow, thymus, and lymph nodes [98,99]. Besides, tiRNAs can be transported to surrounding and distant cells by plasma exosomes [100]. Therefore, it is conceivable that tiRNAs act at sites distant from their production.

Recently, a breakthrough discovery was reported on the role of tiRNA in immunity. This report demonstrated that Schlafen 2 (SLFN2) suppresses the reactive oxygen species (ROS)-induced generation of tiRNA [101]. T cells isolated from mice conditionally depleted of Slfn2 (CD4-Cre *Slfn2^f/f^*) accumulated more oxidative stress-induced tiRNA compared with T cells isolated from control mice. T cells from CD4-Cre *Slfn2^f/f^* mice had impaired antibody, interferon γ (IFNγ), and IL-17A production. The severity of experimental autoimmune encephalomyelitis (EAE) in CD4-Cre *Slfn2^f/f^* mice was much lower compared with that in control mice. These phenomena are explained by the affinity of SLFN2 for tRNAs. SLFN2 binds to tRNAs and protects them from cleavage by ANG [101]. The production of tiRNA is basically regulated by ANG and RNH1 [102,103]. However, at least in T cells, SLFN2 is a new molecule that regulates the production of tiRNA, together with ANG and RNH1 (Figure 2).

One of the most promising strategies used to demonstrate the contribution of tiRNAs to RA pathophysiology is to provoke collagen-induced arthritis (CIA) in CD4-Cre *Slfn2^f/f^* mice. While EAE is a model of multiple sclerosis based on antigens derived from the central nervous system, CIA is a model of RA based on Type II collagen as an autoantigen. Both are animal models of autoimmunity characterized by the activation of antigen-specific T cells, especially Th1 and Th17 [104]. Suppression of EAE in CD4-Cre *Slfn2^f/f^* mice indicated that tiRNAs inhibit the autoimmune response. Therefore, it is expected that tiRNAs also contribute to suppression of the autoimmune response in CIA. This experiment would confirm the potential contribution of tiRNAs to RA pathology during the pre-clinical phase, when autoantibodies such as RF and ACPA are induced.

As the RA phase shifts from pre-clinical to clinical, a lesion called the “pannus” forms in the joints, characterized by synoviocyte proliferation and lymphocyte infiltration [3]. These cells actively synthesize pathogenic cytokines and proteases, contributing to the deterioration of RA. In this context, 5′ tiRNAs with G-rich sequences have the ability to suppress global translation in synoviocytes and lymphocytes, which may reduce cytokine and protease production and suppress RA activity.

In the pannus, excessive protein synthesis, hypoxia, and hypoglycemia are thought to act cooperatively as ER stressors to induce persistent ER stress [105]. To cope with stress, the unfolded protein response (UPR) and ER-related degradation (ERAD) pathways are activated. Failure of these responses causes the release of cytochrome *c* from the mitochondria and subsequent apoptosis [105]. Since tiRNAs protect cells from apoptosis by blocking cytochrome *c* [83], they may suppress the death of synoviocytes and lymphocytes in the pannus under ER stress. In this scenario, tiRNAs are considered as arthritogenic factors that promote pannus formation. Based on the limited findings currently available, it must be said that tiRNAs can have both suppressive and promoting functions in the pathogenesis of RA.

## 6. Conclusions

We discussed the possible role of tiRNAs in RA. Intensive research on tiRNAs as new regulatory ncRNAs has revealed their unique functions, but the analysis of their association with human diseases has just begun. Since inflammatory cytokines are established therapeutic targets for RA, elucidating the association between cytokines and tiRNA activity will prove the importance of tiRNA in the disease. Therefore, it is essential to ask whether the expression of tiRNA regulators such as ANG, RNH1, and SLFN2 is affected by cytokines. We discussed a study that discovered the essential role of tiRNAs in the regulation of lymphocytes. Are tiRNAs also involved in the activation of synoviocytes and osteoclasts, which play pivotal roles in RA progression? By answering these questions, the role of tiRNAs in the pathogenesis of RA will be elucidated. For clinical applications, tiRNAs and related molecules should be carefully investigated for their suitability as therapeutic targets for RA therapy. We hope that tiRNA research will provide clues to identify new biomarkers and design new therapeutic strategies for RA.

## Figures and Tables

**Figure 1 ijms-22-09481-f001:**
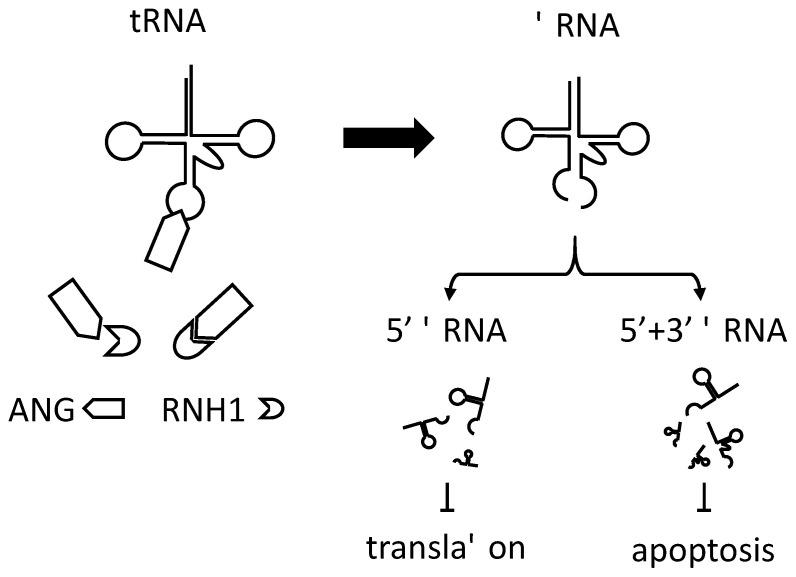
tiRNA production and behavior. Angiogenin (ANG) cleaves a tRNA to produce a tiRNA. ANG is suppressed by ribonuclease inhibitor 1 (RNH1); 5′ tiRNA, but not 3′ tiRNA, is effective in translation suppression. Both 5′ tiRNA and 3′ tiRNA are effective in suppressing apoptosis.

**Figure 2 ijms-22-09481-f002:**
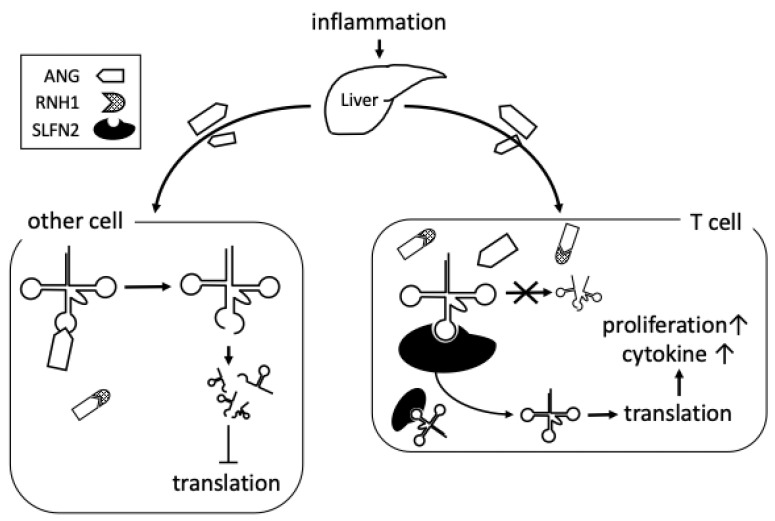
Control of tiRNA in T cells in an inflammatory environment. Inflammation promotes ANG production in the liver. ANG binds to receptors on other cells and is taken up by endocytosis. If incorporated ANG is not inhibited by RNH1, it cleaves mature tRNAs to produce tiRNAs and suppresses global translation. In T cells, SLFN2 protects tRNAs from ROS-mediated cleavage by ANG. This inhibits the production of tiRNA and allows translation for sustained activation of T cells.

## Data Availability

Not applicable.

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
