# Peer review of "Possible Roles of tRNA Fragments, as New Regulatory ncRNAs, in the Pathogenesis of Rheumatoid Arthritis"

_ijms, 2021, doi:10.3390/ijms22179481_

Round 1

Reviewer 1 Report

General comments:

Title of review suggesting role of tRNA fragments in RA pathogenesis. There is still little known about role of tiRNAs in pathogenesis of various diseases therefore this work merits potential interest to readers and may have significant impact for better understanding of role of tRNA fragments. However a clear description of how those molecules is proposed to drive or contribute to pathogenesis of RA is required with more specific examples from the literature is needed. There is not sufficient specific details describing other work from the field in support of the hypothesis that tiRNA take part in RA pathogenesis. Moreover tiRNAs are not the only ones fragments derived from tRNA, there are also tRNA-derived fragment (tRF). tiRNA and tRF differ in the cleavage position of the precursor or mature tRNA transcript. Authors should comment also those type of tRNA derived fragments.

On the other hand in the main text we find a lot of information about other non-coding RNAs not only about tRNA fragments. Some parts of this review seems to be to general or obvious (bulleted in the second part "Specific comments") and other are too short and should be more focused on tiRNAs. In my opinion authors should focus more on tiRANs or change the title of the work.

Specific comments for improvement are provided below.

Unnecessary fragments of review:

  • Line 44, sentence “The second challenge is to identify the cure for RA” is to obvious, I recommended authors avoid using such short,  nothing meaningful wordings
  • historical description sequencing methods (line 72-81) especially part describing DNA sequencing does not add value to the understanding of tiRNAs
  • line 82-88 description of other non-tiRNA databases (as above)
  • line 90-97 basic knowledge, there is no justification for describing the types of RNA in a scientific article whose recipients are people with basic biological knowledge 

In figures 1 and 3 there are word errors: transtalion, inflamation, proliferation: "ti" is missing

Figure 2  The figure does not add much to the review and is unnecessary

Section "conclusions" is too short.

Author Response

We thank the reviewer for the thoughtful comments. We revised our manuscript according to the advice. Followings are point-by-point responses to the reviewer’s comments.

Comment: A clear description of how those molecules is proposed to drive or contribute to pathogenesis of RA is required with more specific examples from the literature is needed. There are not sufficient specific details describing other work from the field in support of the hypothesis that tiRNA take part in RA pathogenesis.

Response: We thanks to the reviewer for valuable advice. Due to the limited research reports on tiRNA on RA, the role of tiRNA in the pathophysiology of RA is largely unknown. We sought to discuss the possibility that tiRNA may be involved in the known pathological mechanism of RA (lines 277 to 301 in chapter 5. Possible roles of tiRNA in Rheumatoid Arthritis in revised manuscript), avoiding excessive speculation.

Comment: TiRNAs are not the only one fragments derived from tRNA, there are also tRNA-derived fragment (tRF). tiRNA and tRF differ in the cleavage position of the precursor or mature tRNA transcript. Authors should comment also those type of tRNA derived fragments.

Response: We agreed to the reviewer’s comment and added information about tRNA fragments other than tiRNA (lines 215 to 226 of the revised manuscript). The word "tRF" seems to refer to several different tRNA fragments in previous reports. To avoid confusion, we classified tRNA fragments into tiRNA and other tRNA fragments, and explained the production pathways for each.

Comment: On the other hand, in the main text we find a lot of information about other non-coding RNAs not only about tRNA fragments. Some parts of this review seem to be to general or obvious (bulleted in the second part "Specific comments") and other are too short and should be more focused on tiRNAs. In my opinion authors should focus more on tiRNAs or change the title of the work.

Response: We have removed the verbose description pointed out by the reviewer. See the answers to the specific comments below. By following the reviewer's thoughtful comments, we believe that the content is more in line with the current title.

Specific comments

Comment: Line 44, sentence “The second challenge is to identify the cure for RA” is too obvious, I recommended authors avoid using such short, nothing meaningful wordings

Response: We agree to the comment and deleted the sentence.

Comment: Historical description sequencing methods (line 72-81) especially part describing DNA sequencing does not add value to the understanding of tiRNAs.

Line 82-88 description of other non-tiRNA databases (as above).

line 90-97 basic knowledge, there is no justification for describing the types of RNA in a scientific article whose recipients are people with basic biological knowledge.

Response: We agree to these comments. We combined lines 72-88 into one sentence and deleted lines 90-97. As a result, we have combined Chapters 2 and 3 into one chapter.

Comment: In figures 1 and 3 there are word errors: transtalion, inflamation, proliferation: "ti" is missing.

Response: We checked the words.

Comment: Figure 2 The figure does not add much to the review and is unnecessary.

Response: We agreed with the comment and deleted Figure 2. Along with this, Fig. 3 has been changed to Fig. 2.

Comment: Section "conclusions" is too short.

Response: We have rewritten the conclusion. We have proposed some examples of studies to clarify the function of tiRNA in rheumatoid arthritis.

Reviewer 2 Report

In this manuscript, the authors provide an informative and elegantly written overview on non-coding RNA’s in general, with a focus on the principles of tRNA’s and tiRNA’s. This review is motivating and likely to attract more attention to this exiting research topic.

I have the following comments:

Major: Following the titles of my own brief literature search, I retrieved some potentially relevant papers, which are not mentioned in this manuscript. I suggest to briefly and critically discuss all the present (still very limited) evidence for a role of ncRNA’s from papers on articular tissues and cells in RA or arthritis models. 

Minor: I think that there is a spelling error in Line 168: translation instead of translational

Author Response

We thank the reviewer for the important comments. We revised our manuscript according to the advice. Followings are point-by-point responses to the reviewer’s comments.

Major Comment: Following the titles of my own brief literature search, I retrieved some potentially relevant papers, which are not mentioned in this manuscript. I suggest to briefly and critically discuss all the present (still very limited) evidence for a role of ncRNA’s from papers on articular tissues and cells in RA or arthritis models.

Response: We consider this comment to be very important and reaffirmed that there are some limitations to ncRNA research. In particular, miRNAs have an off-target effect. This makes it difficult to identify the definitive miRNAs involved in the pathology of RA, and many problems must be overcome before clinical application. We referred to the typical problem of miRNA research in RA and briefly discussed this in lines 103-111 of the revised manuscript.

Minor Comment: I think that there is a spelling error in Line 168: translation instead of translational.

Response: Thank you for pointing out the misspelling. We corrected it in the revised manuscript.

Round 2

Reviewer 1 Report

Revised manuscript requires thorough editorial corrections.

Please check the text carefully after editing - some parts of the text are repeated (example Line 72 is triplicated: the same is line 100-108 and 110-119  or line 293-301 is the same as 304-312 or 343-352 duplicated in lines 364-369 etc.)

In figures 1 there is still word error: in transtalion: "ti" is missing

Line 406: how research on tiRNAs might provide new diagnostic techniques? whether instead of "diagnostic techniques" there should be “biomarkers”?

Author Response

Comments: Revised manuscript requires thorough editorial corrections.

Response: Once again, our revised manuscript has undergone another English language editing by MDPI.

Comments: Please check the text carefully after editing - some parts of the text are repeated (example Line 72 is triplicated: the same is line 100-108 and 110-119  or line 293-301 is the same as 304-312 or 343-352 duplicated in lines 364-369 etc.). Comments:  In figures 1 there is still word error: in transtalion: "ti" is missing.

Response: The revised manuscript has undergone another English compilation by MDPI. To tell the truth, we couldn't find these mistakes. We were afraid that a conversion error might have occurred, so we also contacted the editorial office on these issues.

Comments: Line 406: how research on tiRNAs might provide new diagnostic techniques? whether instead of "diagnostic techniques" there should be “biomarkers”?

Response: Thank you for your very good advice. We found that "biomarker" is exactly what we thought it would be, so we use "biomarker" in the revised manuscript.

We can see that this review has improved a lot thanks to the reviewer's comments. We would like to express our respect and gratitude to the reviewers.

Round 3

Reviewer 1 Report

"To tell the truth, we couldn't find these mistakes. We were afraid that a conversion error might have occurred, so we also contacted the editorial office on these issues."

I am enclosing a screenshot of the version of the manuscript that I can see with the error I mentioned marked in red line. I do not understand why I still see editorial errors, so the decision to accept the mannuscript should be made by the editor.
